# Mortality in patients with adrenal insufficiency: a protocol for a systematic review and meta-analysis

Francesca Allosso,[1] Konstantinos Dalakas,[2,3] Ragnhildur Bergthorsdottir,[2,3] Dimitrios Chantzichristos [ID] ,[2,3] Eva Hessman,[4] Bright I Nwaru,[5,6] Emanuele Bobbio,[7,8] Daniela Pasquali,[1] Gudmundur Johannsson [ID] ,[2,3] Daniela Esposito [ID] [2,3]

FA and KD contributed equally.

For numbered affiliations see end of article.

**Correspondence to**
Dr Daniela Esposito;
daniela.esposito@gu.se

## ABSTRACT

**Introduction** Adrenal insufficiency (AI) is a rare disorder characterised by an impaired secretion of glucocorticoids from the adrenal glands. Treatment strategies for AI have developed over time with reduced glucocorticoid replacement doses and improved circadian exposure regimens, but whether this has resulted in better survival is unknown. The main purpose of this systematic review is to gather and synthesise available evidence on long-term mortality in patients with AI. The secondary aim is to study causes of death, with focus on cardiovascular and infectious diseases, in AI patients.

**Methods and analysis** Studies published from the inception of respective databases (Medline, Embase, Cochrane and Web of Science) until the end of May 2023 will be systematically synthesised. Observational studies with a reference population will be included, and their quality will be assessed using the Newcastle-Ottawa scale. Data collected will be narratively integrated and a meta-analysis will be performed to pool data from studies considered homogeneous. The systematic review will be reported following the Preferred Reporting Items for Systematic Reviews and Meta-Analyses guidelines. This will be the first systematic review assessing mortality and causes of death in AI patients. The findings of this systematic review will be of value for both patients and healthcare providers.

**Ethics and dissemination** This systematic review does not require ethical approval or informed consent because it will be based on previously published data only and does not implicate any direct contact with individual patients. The research results will be presented at scientific conferences and submitted for publication in an internationally recognised peer-reviewed scientific journal.

**PROSPERO registration number** CRD42023416253.

## STRENGTHS AND LIMITATIONS OF THIS STUDY

⇒ This systematic review will for the first time provide a comprehensive synthesis of overall mortality in patients with adrenal insufficiency.
⇒ The protocol of this systematic review follows the Preferred Reporting Items for Systematic Review and Meta-Analysis Protocols guidelines.
⇒ A detailed description of the review processes is presented in this protocol increasing the reproducibility of the systematic review.
⇒ All available studies on mortality in adrenal insufficiency will be gathered from publicly available literature databases, without language or geographical limitations.
⇒ A potential limitation can be high heterogeneity in the population, as both primary and secondary adrenal insufficiency are going to be included. However, separate subgroup analyses will be performed.

## INTRODUCTION

Adrenal insufficiency (AI) is a rare disorder characterised by an impaired secretion of glucocorticoids (GC) from the adrenal cortex.[1] Primary AI (PAI), with an estimated prevalence of 93–144/million in Europe, is caused by disorders of the adrenal glands and is usually associated with mineralocorticoid and androgen deficiency.[1] It can be acquired or inherited as in congenital adrenal hyperplasia (CAH), usually due to steroid 21-hydroxylase deficiency.[2] In western countries, acquired PAI is most commonly due to an autoimmune-mediated adrenalitis, which can be isolated or combined with other autoimmune diseases (eg, autoimmune polyglandular syndrome type 1 or 2).[2,3] Other causes of PAI are tuberculosis and infectious diseases.[3] Secondary AI (SAI), with an estimated prevalence of 150–280/million, is caused by adrenocorticotropic hormone (ACTH) deficiency secondary to a pituitary disease and is often associated with other pituitary hormone deficiencies.[1,2]

Treatment of AI consists of replacement therapy with GC.[1] In PAI, mineralocorticoid replacement therapy is also required while in SAI additional replacement therapy with thyroid hormone, sex steroids and growth hormone may be needed due to failure of other pituitary hormone axes.[2] Before

the introduction of GC therapy, most patients with PAI died within 2 years of diagnosis.[4] Although long-term outcomes have markedly improved, mortality remains increased, with cardiovascular disease being the leading cause of death.[5–7]

Standard replacement treatments in AI include short-acting GC such as hydrocortisone and cortisone acetate, usually given 2–3 times daily.[8] In some countries, other GCs such as prednisolone are used.[2] However, conventional GC replacement fails to restore a circadian cortisol profile in patients with AI.

There have been attempts to improve the dosing regimen of GC replacement therapy and the management of intercurrent illnesses.[9] The introduction of new therapies, as once-daily modified release hydrocortisone that better mimics the endogen circadian production of cortisol, has been associated with a more favourable metabolic outcome.[10] Whether this has led to improvement in survival rate is however unknown. Data on mortality in patients with AI are discordant.[6 11–14] Despite having GC deficiency in common, PAI and SAI have different aetiologies. SAI is usually associated with other pituitary deficiencies such as growth hormone deficiency, which per se could lead to increased risk of cardiovascular diseases and death.[15] On the other hand, PAI is often associated with other autoimmune diseases, which may play an important role on patients' outcome.[16]

No previous studies have so far synthesised the available evidence on mortality in patients with AI. In this systematic review, we will analyse overall and disease-specific mortality in patients with PAI and SAI.

### Study aims and objectives

The primary aim of this systematic review is to identify, critically appraise and synthesise available data on long-term mortality in adult patients with AI. Secondary aims include the assessment of mortality due to cardiovascular and infectious diseases.

## METHODS AND ANALYSIS

The protocol for this review follows the recommendation of the Preferred Reporting Items for Systematic Review and Meta-Analysis Protocols (PRISMA) statement and has been registered in PROSPERO with registration number: CRD42023416253.[17]

This systematic review was planned in November 2022 and is expected to be finalised in June 2024. In the final systematic review, we will describe any deviation from the protocol. The final systematic review will be reported according to the PRISMA guidelines.[17]

### Eligibility criteria

The population of interest will be adult patients with AI, and this includes patients with PAI (including CAH) and SAI. Studies reporting long-term mortality (defined as death occurring ≥1 year from the diagnosis of AI) will be eligible for inclusion. Data across different age groups,

gender and ethnicity will be analysed, and there will be no exclusion by country or language. Patients with Waterhouse-Friderichsen syndrome, leucodystrophy and type 4 renal tubular acidosis will be excluded from the analysis. Studies not including a comparison group (ie, a background population) will be excluded. In addition, review articles, non-peer reviewed articles, commentaries, proceedings, laboratory science studies, case studies, case series and other studies that do not allow the calculation of rates of the outcomes will be excluded from this systematic review. If an eligible study does not report the incidence of the prespecified outcomes, we will attempt to contact the study authors to request for the respective details.

### Information sources

Our data sources will include electronic databases, conference abstracts, grey literature, and researchers and authors themselves. A systematic search will be performed through Medline, Embase, Cochrane and Web of Science using commonly used phrases stated in related literature, in addition to controlled vocabulary, that is, Medical Subject Headings (MeSH) terms in Ovid Medline and Cochrane Central and Emtree terms in Embase. The databases will be searched from their inception date until the end of May 2023. Broad and inclusive search terms will be used in order to limit the risk of omitting any relevant studies.

We will review reference lists and citations of included studies in order to find additional relevant articles. In addition, we will perform an updated search at the completion of the systematic review to ensure that studies published after the first search will be included.

### Search strategy

Our initial search syntax for Ovid Medline is given below and this will be adapted in searching the other databases.

1. Adrenal Insufficiency/ or Addison Disease/ or Hypoadrenocorticism, Familial/ or Adrenoleukodystrophy/ or Hypoaldosteronism/ or Waterhouse-Friedrichsen syndrome/ or Hypopituitarism/ or Adrenal Hyperplasia, Congenital/
2. ('Adrenal Insufficiency' or Hypoadrenalism or 'Adrenal Gland Hypofunction' or Addison* or 'Adrenocortical Insufficiency' or 'Familial Hypoadrenocorticism' or Adrenoleukodystrophy or Hypoaldosteronism or 'Waterhouse-Friedrichsen syndrome' or 'Complex Glycerol Kinase Deficiency' or 'Xp21 Contiguous Gene Deletion Syndrome' or 'X-linked Adrenal Hypoplasia' or 'X-Linked Addison Disease' or 'Cytomegalic Adrenocortical Hypoplasia' or 'Congenital Adrenal Hypoplasia' or 'Type IV Renal Tubular Acidosis' or 'Purpura Fulminans' or 'Meningococcal Hemorrhagic Adrenalitis' or Hypopituitarism or 'Congenital Adrenal Hyperplasia').tw,kf.
3. (mortality or death* or ((fatal or long-term) adj2 outcome*)).tw,kf. /freq=2

4. Mortality/ or Death/ or Survival/ or Fatal Outcome/ or Mortality.fs.
5. (exp animal/ or exp invertebrate/ or animal experiment/ or animal model/ or exp plant/ or exp fungus/) not exp human/
6. 1 or 2
7. 3 or 4
8. 6 and 7
9. 8 not 5

## Selection process

Articles will be independently screened by two reviewers using the Rayyan web application for systematic reviews. First, articles will be screened by titles and/or abstracts. The screened articles will be classified into three groups: relevant, irrelevant and uncertain. Articles that both reviewers consider to be irrelevant will be excluded from the review. For articles that are considered relevant or uncertain based on their title and/or abstract, the full-text article will be retrieved to assess eligibility. Potentially relevant articles selected by at least one author will be investigated and evaluated in full. Next, the full text of the eligible articles will be reviewed and a list of articles to be included by each reviewer will be created. The two lists will then be compared, and the discrepancy will be discussed. If no agreement is reached, the entire team will make the final decision. A flow diagram showing the selection process will be created using the PRISMA guidelines.

## Data extraction

Predefined information of interest from included studies will be extracted using a data extraction form. This will help standardise the information recorded and aid analyses. The following data will be extracted independently by two reviewers: study name (along with the name of the first author and year of publication), country where the study was conducted, study size, source from which patients or study participants were selected, study design, definition of AI, definition of CAH, disease duration, outcomes definition, age, gender, ethnicity, incidence or prevalence with 95% CIs or the number of patients achieving the outcomes. Data on causes of PAI and SAI will be systematically gathered, as underlying conditions themselves may independently influence the outcomes. Data on doses of GC and Mineralocorticoids (MC) will be collected, as treatment strategies have changed over time. Studies including patients with hypopituitarism will be included with the aim to gather data on patients with SAI. If the required data are inadequate, unclear or missing from the article, authors will be contacted. If needed, missing information will be calculated from the available data when possible. In case of studies with overlapping population, only the study with the largest sample size and higher quality will be included. The data extraction will be discussed by two reviewers, and the entire team will arbitrate in the case of any disagreement.

## Quality assessment

The quality of the studies will be independently assessed by two authors using the Newcastle-Ottawa quality assessment scale. Critical appraisal of articles will be performed to assess the methodological quality of each article. The quality appraisal will be undertaken by two reviewers and a third reviewer will arbitrate in the case of any disagreement.

## Data synthesis

Characteristics and main results of the included studies will be presented in descriptive tables and the gathered evidence will be synthesised narratively.

We will calculate rate of mortality in each study if possible. If the included studies are sufficiently homogeneous, we will undertake a meta-analysis by pooling estimates across studies using the random effects approach.

Heterogeneity between studies will be assessed using the Higgins $I^2$ statistic.[18] $I^2$ values of 0%–30% will be considered as minimal heterogeneity, 31%–50% moderate heterogeneity and> 50% substantial heterogeneity.[18] As PAI and SAI are two different entities, we will conduct separate subanalyses on mortality in PAI and SAI. If data are sufficient, we will conduct subgroup analysis to explore potential reasons for heterogeneity based on the quality of study, country, age, gender and ethnicity. A subgroup analysis will be also conducted for patients with CAH.

Begg's funnel plot and Egger's test will be used to assess publication bias.[19] The former is a scatter plot of the log ORs of individual studies on the x-axis against 1/SE of each study on the y-axis.[19] Egger's test is a linear regression test of the normalised effect estimate (log OR/SE) against its precision (1/SE).[19] A p<0.10 on Egger's test or an asymmetrical funnel plot will be considered to indicate the presence of publication bias.

## ETHICS AND DISSEMINATION

This systematic review does not require ethical approval or informed consent because it will be based on previously published data only and does not implicate any direct contact with individual patients. The research results will be presented at scientific conferences and submitted for publication in an internationally recognised peer-reviewed scientific journal.

## PATIENT AND PUBLIC INVOLVEMENT

No patient will be involved in this systematic review.

## DISCUSSION

AI is a life-threatening disorder. In the past, most patients with AI died within a few years after diagnosis due to scarce treatment options.[3] Thanks to the discovery of cortisone in the 1940s, the survival of patients with AI has improved.[3] In the 1960s, Mason *et al* showed that PAI

patients on GC replacement therapy had a mortality rate similar to that of the background population.[20] In line with that study, a recent Norwegian study, including 811 patients with PAI, showed no difference in mortality rate between PAI patients and the general Norwegian population.[11] Conversely, our group showed that mortality in PAI patients was twofold increased in comparison with the general Swedish population, after following 1675 patients with PAI between 1987 and 2001.[6] More recent data from UK have confirmed this finding showing a more than twofold excess mortality in patients with PAI.[16]

Mortality in AI patients has been related to cardiovascular diseases, infections, acute adrenal crises and malignancies.[7] In a large retrospective cohort study including 6821 patients with AI from the UK, Ngaosuwan *et al* have recently shown that overall mortality is higher by approximately 80% in PAI and 50% in SAI in comparison with matched controls.[7] Another study on the same cohort showed that mortality due to cardiovascular disease was increased in both PAI and SAI compared with matched controls.[21] Interestingly, patients with SAI had an increased risk of cerebrovascular events when compared with PAI patients.[21]

Other common causes of death in AI are infectious diseases.[7] A multicentre study including 2034 patients with PAI and SAI from the European Adrenal Insufficiency Registry showed that the major causes of death were cardiovascular disease (35%) and infectious diseases (15%).[22] In a population-based study, including 3299 patients with PAI, Bensing *et al* have found that the risk of death due to infections was sixfold increased compared with the general population.[5] Moreover, in a cohort study of 1286 hypopituitary patients, Burman *et al* have found that mortality from infections was increased in ACTH-deficient patients but not in patients without ACTH deficiency.[12]

Inappropriate replacement therapy with GCs has been described as the main determinant of a poor prognosis in AI patients.[23] Excessive cortisol exposure and GC replacement therapy that does not mimic cortisol circadian rhythm are indeed associated with the adverse metabolic profile, increased morbidity and mortality.[23 24]

Over the last two decades, efforts have been made to optimise GC treatment, trying to reduce the total daily GC dose and optimise intercurrent illness management.[9] In addition, novel treatment strategies have been introduced to better mimic the physiological diurnal variation in cortisol levels.[10] Whether this has led to an improvement in long-term outcomes remains unknown. Previous studies conducted on this topic have shown controversial findings due to different study designs, populations and settings.[5 6 11] Since AI is a rare disease, no single centre has accumulated sufficient experience to draw definitive conclusions. Therefore, a systematic review is needed to provide clearer insights into this topic. Increasing knowledge of AI mortality can lead to optimised management of AI and its comorbidities.

**Author affiliations**
¹Department of Advanced Medical and Surgical Sciences, University of Campania Luigi Vanvitelli, Naples, Italy
²Department of Internal Medicine and Clinical Nutrition, Institute of Medicine, Sahlgrenska Academy, University of Gothenburg, Gothenburg, Sweden
³Department of Endocrinology, Sahlgrenska University Hospital, Gothenburg, Sweden
⁴Biomedical Library, Gothenburg University Library, University of Gothenburg, Gothenburg, Sweden
⁵Krefting Research Centre, Institute of Medicine, Sahlgrenska Academy, University of Gothenburg, Gothenburg, Sweden
⁶Wallenberg Centre for Molecular and Translational Medicine, Institute of Medicine, Sahlgrenska Academy, University of Gothenburg, Gothenburg, Sweden
⁷Department of Cardiology, Sahlgrenska University Hospital, Gothenburg, Sweden
⁸University of Gothenburg, Gothenburg, Sweden

**Contributors** DE conceived this study. FA, KD, DP and DE developed the study protocol. All the authors revised the study protocol. FA and KD will implement the systematic review under the supervision of DE, EB, RB, DC, BIN, DP and GJ. EH will develop the search strings and conduct the study search. EB and BIN will provide the statistical analysis expertise. EB, BIN and DE will conduct data analysis. FA and KD will perform the screening and extraction of data whereas DE, EB, RB, DC, BIN and GJ will review the work.

**Funding** This work received support from the Swedish state under the agreement between the Swedish government and the county councils, the ALF-agreement (ALFGBG-874631 and ALFGBG-960884).

**Competing interests** None declared.

**Patient and public involvement** Patients and/or the public were not involved in the design, or conduct, or reporting, or dissemination plans of this research.

**Patient consent for publication** Not applicable.

**Provenance and peer review** Not commissioned; externally peer reviewed.

**ORCID iDs**
Dimitrios Chantzichristos http://orcid.org/0000-0002-1660-1973
Gudmundur Johannsson http://orcid.org/0000-0003-3484-8440
Daniela Esposito http://orcid.org/0000-0001-8993-2071

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
