## [Reviewer comments · BMJ Open]

ARTICLE DETAILS

TITLE (PROVISIONAL)	Mortality in patients with adrenal insufficiency: a protocol for a systematic review and meta-analysis
AUTHORS	Alloso, Francesca; Dalakas, Konstantinos; Bergthorsdottir, Ragnhildur; Chantzichristos, Dimitrios; Hessman, Eva; Nwaru, Bright; Bobbio, Emanuele; Pasquali, Daniela; Johannsson, Gudmundur; Esposito, Daniela

VERSION 1 – REVIEW

REVIEWER	Stewart Albert Saint Louis Regional Medical Center
REVIEW RETURNED	26-Jun-2023

GENERAL COMMENTS	The authors have proposed a meta-analysis to evaluate the effect of adrenal insufficiency on mortality. In their work sheet: They need to define methods of definition of adrenal insufficiency per article. They need to define acute effects (ICU mortality, in hospital mortality, 30-day mortality) For Waterhouse Frederickson and meningitis they need to describe an appropriate control population , as both conditions are associated with high rates of mortality independent of AI For leukodystrophy they need to define their population and controls as AI will be found in more advanced neurological disease and in older males For congenital adrenal hyperplasia they need to define the severity of the disease as many patients have partial CAH Type 4 RTA is not necessarily associated with AI For secondary AI, they need to separate acute mortality –eg surgery from chronic effects For secondary AI they need to define congenital vs surgical vs radiation vs immunotherapy vs cancer For sepsis/ICU induced AI they need to define methods of definition of AI vs therapy with steroids Studies should be analyzed/grouped as acute series vs long term or
---

	epidemiological series Era of treatment should be evaluated – doses of glucocorticoids and mineralocorticoids have changed over time How will they define cardiovascular mortality vs infectious complications?
--	--

REVIEWER	KANCHANA NGAOSUWAN Chulabhorn Royal Academy
REVIEW RETURNED	03-Jul-2023

GENERAL COMMENTS	The study protocol is generally well-written, with clear research questions and organized search terms and strategy. However, there are concerns about the methodology.  • It is hypothesized that the quality of care for AI patients has improved over time, potentially leading to a reduction in mortality rates among these patients. In order to investigate this speculation, the author plans to conduct a time trend analysis. The author stated that “to analyse time trends in mortality in AI patients a subgroup analysis will be performed based on time during which the studies were conducted”. This statement needs clarification because most studies on AI are retrospective and span over a decade or longer to ensure statistical validity, given the rarity of the disorder. Despite including AI patients over a long period, many studies did not distinguish the mortality rates AI according to the different time periods. Furthermore, almost all previous studies did not compare the mortality rates of AI patients with those of the reference population living in the same time period. It is unclear how the author plans to synthesize this evidence. • Another concern is the definition of secondary AI in this protocol. Numerous studies have reported mortality rates, including cardiovascular mortality, in patients with hypopituitarism. However, the majority of these patients have growth hormone deficiency and may not necessarily have secondary AI. Will these patients be included in the analysis? • The previous literature has reported the mortality risk of AI patients using two different approaches. The first approach involved calculating the "standardized mortality ratio" by comparing the mortality rate of AI patients to the expected death rates obtained from national statistics. In more recent studies, the mortality risk was reported using the "Hazard ratio" by comparing the mortality rate of AI patients to matched controls. In the meta-analysis, it is unclear how these different sources of data will be handled and synthesized.
--

VERSION 1 – AUTHOR RESPONSE

Comments from reviewer #1

1. They need to define methods of definition of adrenal insufficiency per article.

Answer: We agree with the reviewer that this is an important information that will help us to assess the quality of the included articles. Information on the definition of adrenal insufficiency will be collected from the included articles and shown in the systematic review. This information has now been added in the revised version of the protocol. Please, see the highlighted changes on page 9, line 206.

2. They need to define acute effects (ICU mortality, in hospital mortality, 30-day mortality).

Answer: Thank you for drawing attention to this point. The aim of this systematic review is to synthesize available data on long-term mortality (defined as death at ≥ 1 year from the diagnosis of

adrenal insufficiency). We will not evaluate acute mortality, because the causes of death may be diverse and possibly not related to adrenal insufficiency but to the underlying intercurrent condition (e.g. sepsis, trauma etc.). This would result in high heterogeneity between studies. Therefore, only articles assessing long-term overall mortality and cause-specific mortality will be included in our study. This has been clarified in the protocol. Please, see the highlighted changes on page 6, lines 122 and 135-136.

3. For Waterhouse-Friderichsen and meningitis they need to describe an appropriate control population, as both conditions are associated with high rates of mortality independent of AI.

4. For leukodystrophy they need to define their population and controls as AI will be found in more advanced neurological disease and in older males.

Answer to comments 3 and 4: We agree with the reviewer that patients with Waterhouse-Friderichsen syndrome and meningitis as well as leukodystrophy may have an increased mortality regardless of AI. Therefore, these patients will be excluded in the systematic review.

Waterhouse-Friderichsen, meningitis and leukodystrophy were however, used as key words in the search strategy because adrenal insufficiency is a rare entity, and the aim was to use broad search criteria in order to avoid missing any relevant articles. We acknowledge that this information was not clear in the protocol, and it has been addressed in the revised version. Please, see the highlighted changes on pages 6 and 7, lines 138-140 and lines 154-155.

5. For congenital adrenal hyperplasia they need to define the severity of the disease as many patients have partial CAH.

Answer: The authors agree with this comment as patients with different forms of CAH (e.g. as salt wasting, non-classical etc.) have different long-term outcomes. This information will be collected from the included articles with CAH and will be documented in the systematic review. In addition, a separate analysis of mortality will be performed for patients with CAH. Please, see the highlighted changes on the protocol on page 9, line 206, and on page 10, lines 238-239.

6. Type 4 RTA is not necessarily associated with AI.

Answer: The authors agree with this comment. The reason why this disease is included in the search strategy is our approach towards broad search criteria. This patient group is going to be excluded from the meta-analysis. This has now been clarified in the protocol. Please, see the highlighted changes on pages 6-7, lines 139-140.

7. For secondary AI, they need to separate acute mortality –eg surgery from chronic effects.

Answer: The aim of this systematic review is to synthesize available data on long-term mortality (defined as death at ≥ 1 year from the diagnosis of adrenal insufficiency). For this reason, only long-term mortality is going to be analysed. We acknowledge that this was not clearly stated in the protocol, and it has now been clarified (Please, see also answer to comment n.2). Please, see the highlighted changes on page 6, lines 122 and 135-136.

8. For secondary AI they need to define congenital vs surgical vs radiation vs immunotherapy vs cancer.

Answer: We agree with this point regarding the significance of distinguishing between various causes of secondary adrenal insufficiency, including congenital, surgical, radiation, immunotherapy, and cancer-related aetiologies. Indeed, this differentiation is of importance, as these distinct causes may have a different impact on long-term mortality. Therefore, these data will be extracted from the included articles and shown in the systematic review. This has now been stated clearly in the protocol. Please, see the highlighted changes on page 9, lines 208-209.

9. For sepsis/ICU induced AI they need to define methods of definition of AI vs therapy with steroids.

10. Studies should be analyzed/grouped as acute series vs long term or epidemiological series.

Answer to comments 9 and 10: We have now clarified in the protocol that only studies with long-term mortality will be included (please see answer to comments 2 and 7). Please, see the highlighted changes on page 6, lines 122 and 135-136.

11. Era of treatment should be evaluated – doses of glucocorticoids and mineralocorticoids have changed over time.

Answer: The authors agree that this is an important information, as through the years there has been a tendency to be more restrictive with the GC doses. Hence, data on treatment strategies and doses with glucocorticoids and mineralocorticoids will be extracted from the included articles, as suggested. The protocol has been adjusted accordingly. Please, see the highlighted changes on page 9, lines 210-211.

12. How will they define cardiovascular mortality vs infectious complications?

Answer: Thank you for question. The definition of cardiovascular and infectious diseases is predefined by the studies. Data on cardiovascular mortality and infectious diseases definition will be collected from the included articles during the data extraction. Specifically, for registry studies, specific international diagnostic codes (ICD) used in the article will be collected. In clinical studies detailed descriptive criteria will be gathered. This approach will also help us to assess the quality of the included articles. Please, see the highlighted changes on page 9, line 206.

Comments from reviewer #2

1. It is hypothesized that the quality of care for AI patients has improved over time, potentially leading to a reduction in mortality rates among these patients. In order to investigate this speculation, the author plans to conduct a time trend analysis. The author stated that “to analyse time trends in mortality in AI patients a subgroup analysis will be performed based on time during which the studies were conducted”. This statement needs clarification because most studies on AI are retrospective and span over a decade or longer to ensure statistical validity, given the rarity of the disorder. Despite including AI patients over a long period, many studies did not distinguish the mortality rates AI according to the different time periods. Furthermore, almost all previous studies did not compare the mortality rates of AI patients with those of the reference population living in the same time period. It is unclear how the author plans to synthesize this evidence.

Answer: The authors agree with the reviewer there are some challenges in the analysis of time trends in mortality. To address these issues, we intend to collect detailed data on the study period from each included article. We agree on the importance of comparing the mortality rates of AI patients with the reference population living during the same time period. Therefore, if any studies fail to provide this important information, it will be excluded from the analysis. If data on that issue are limited, to enhance the reliability of our results, we will consider performing a separate analysis for studies using Standardized Mortality Ratios (SMR), a method that takes into account gender, age, and calendar year matching with the general population. Please, see the highlighted changes on page 10, lines 239-243.

2. Another concern is the definition of secondary AI in this protocol. Numerous studies have reported mortality rates, including cardiovascular mortality, in patients with hypopituitarism. However, the majority of these patients have growth hormone deficiency and may not necessarily have secondary AI. Will these patients be included in the analysis?

Answer: Thank you for bringing up this issue, as it was not specified in the protocol previously. Our aim is to include these articles and retrieve information in the subgroup of patients with secondary AI. If these data are not available, the authors of the respective studies will be contacted to retrieve the information required. With this strategy we are aiming to secure that patients included in the analysis have secondary AI. This issue has now been clarified in the protocol. Please, see the highlighted changes on page 9, lines 211-212.

3. The previous literature has reported the mortality risk of AI patients using two different approaches. The first approach involved calculating the "standardized mortality ratio" by comparing the mortality rate of AI patients to the expected death rates obtained from national statistics. In more recent studies, the mortality risk was reported using the "Hazard ratio" by comparing the mortality rate of AI patients to matched controls. In the meta-analysis, it is unclear how these different sources of data will be handled and synthesized.

Answer: Thank you for feedback on this important aspect of the statistical analysis. The authors agree that there can be heterogeneity among the statistical analyses of different studies. Although SMR and HR can be comparable, it is important to recognize a fundamental difference. In studies using HR, there is often adjustment for several potential confounding variables in the regression model, whereas SMR standardization typically involves only a limited set of confounders, such as sex, age, and calendar year. Therefore, we will conduct separate meta-analyses for studies reporting SMR and those reporting HR. This approach will ensure a high reliability and accuracy of the synthesis of evidence.

VERSION 2 – REVIEW

REVIEWER	Stewart Albert Saint Louis Regional Medical Center
REVIEW RETURNED	21-Nov-2023

GENERAL COMMENTS	Accept
--------

REVIEWER	KANCHANA NGAOSUWAN Chulabhorn Royal Academy
REVIEW RETURNED	30-Nov-2023

GENERAL COMMENTS	Thank you for the author's response to my comments. I understood all points except one. Regarding the time trend analysis, the inclusion of studies comparing the mortality rates of AI patients with those of controls (or the general population) living in the same calendar years does not adequately address this issue. Each study provided only a single estimate of mortality risk (e.g., HR, SMR, or overall mortality rates), likely representing the "average" estimate of mortality risk over a long period of time (i.e., decades). Consequently, it is still unclear which statistical method of meta-analysis the author intends to use in evaluating the changing trend of mortality risk over time.
--

VERSION 2 – AUTHOR RESPONSE

Comments from reviewer #2

1. Thank you for the author's response to my comments. I understood all points except one.

Regarding the time trend analysis, the inclusion of studies comparing the mortality rates of AI patients with those of controls (or the general population) living in the same calendar years does not adequately address this issue. Each study provided only a single estimate of mortality risk (e.g., HR, SMR, or overall mortality rates), likely representing the "average" estimate of mortality risk over a long period of time (i.e., decades). Consequently, it is still unclear which statistical method of meta-analysis the author intends to use in evaluating the changing trend of mortality risk over time.

Answer: The authors agree with the reviewer's observation that analyzing time trends in mortality poses a considerable challenge. Initially, the plan was to explore whether mortality in AI patients has change over time by stratifying the analysis into distinct time periods (e.g., mortality in studies including patients before 2000 versus those recruiting patients after 2000). Similar approaches have been undertaken in other meta-analyses examining mortality in rare endocrine diseases, as evidenced by:

- Bolfi et al.; European Journal of Endocrinology, 2018; 179, Pages 59–71, doi: 10.1530/EJE-18-0255
- Dekkers et al.; J Clin Endocrinol Metab. 2008;93(1):61-7. doi: 10.1210/jc.2007-1191

However, as pointed out by the reviewer, the majority of studies on mortality in AI are retrospective and span several decades due to the rarity of the disorder. The team has extensively discussed this issue and acknowledges that conducting such an analysis may not be feasible due to the limited number of studies available on the topic. Consequently, the decision has been made to exclude this analysis from the protocol. We appreciate the reviewer for bringing this concern to our attention.

Please, see the highlighted changes in the revised version of the manuscript.

VERSION 3 – REVIEW

REVIEWER	KANCHANA NGAOSUWAN Chulabhorn Royal Academy
REVIEW RETURNED	23-Dec-2023
GENERAL COMMENTS	Many thanks to the author team for their diligent approach to the methodology.